# The Antitumoral Effect In Ovo of a New Inclusion Complex from Dimethoxycurcumin with Magnesium and Beta-Cyclodextrin

**DOI:** 10.3390/ijms25084380

**Published:** 2024-04-16

**Authors:** Marco A. Obregón-Mendoza, William Meza-Morales, Karla Daniela Rodríguez-Hernández, M. Mirian Estévez-Carmona, Leidys L. Pérez-González, Rosario Tavera-Hernández, María Teresa Ramírez-Apan, David Barrera-Hernández, Mitzi García-Olivares, Brian Monroy-Torres, Antonio Nieto-Camacho, María Isabel Chávez, Rubén Sánchez-Obregón, Raúl G. Enríquez

**Affiliations:** 1Instituto de Química, Universidad Nacional Autónoma de México, Mexico City 04510, Mexico; marco.obregon@zaragoza.unam.mx (M.A.O.-M.); william.meza@upr.edu (W.M.-M.); xandy411@comunidad.unam.mx (K.D.R.-H.); leidyslaura92@gmail.com (L.L.P.-G.); rosario.tavera@gmail.com (R.T.-H.); mtrapan@yahoo.com.mx (M.T.R.-A.); brianuami@gmail.com (B.M.-T.); camanico2015@yahoo.com (A.N.-C.); isabel@unam.mx (M.I.C.); rubens@unam.mx (R.S.-O.); 2Departamento de Farmacia, Escuela Nacional de Ciencias Biológicas, Instituto Politécnico Nacional, M. Wilfrido Massieu SN, U. A. Zacatenco, Mexico City 07738, Mexico; mirianestevezc@gmail.com; 3Departamento de Biología de la Reproducción “Dr. Carlos Gual Castro”, Instituto Nacional de Ciencias Médicas y Nutrición Salvador Zubirán, Mexico City 14080, Mexico; barrera1912@gmail.com (D.B.-H.); mariol_48@ciencias.unam.mx (M.G.-O.)

**Keywords:** curcuminoids, 3,4-dimethoxycurcumin, homoleptic complexes, inclusion complexes, beta-cyclodextrin complexes, antitumoral activity, breast cancer

## Abstract

Breast cancer is one of the leading causes of death in the female population because of the resistance of cancer cells to many anticancer drugs used. Curcumin has cytotoxic activities against breast cancer cells, although it has limited use due to its poor bioavailability and rapid metabolic elimination. The synthesis of metal complexes of curcumin and curcuminoids is a relevant topic in the search for more active and selective derivatives of these molecular scaffolds. However, solubility and bioavailability are concomitant disadvantages of these types of molecules. To overcome such drawbacks, the preparation of inclusion complexes offers a chemical and pharmacologically safe option for improving the aqueous solubility of organic molecules. Herein, we describe the preparation of the inclusion complex of dimethoxycurcumin magnesium complex (DiMeOC-Mg, (**4**)) with beta-cyclodextrin (DiMeOC-Mg-BCD, (**5**)) in the stoichiometric relationship 1:1. This new inclusion complex’s solubility in aqueous media phosphate buffer saline (PBS) was improved by a factor of 6x over the free metal complex (**4**). Furthermore, **5** affects cell metabolic rate, cell morphology, cell migration, induced apoptosis, and downregulation of the matrix metalloproteinase-2 (MMP-2) and matrix metalloproteinase-9 (MMP-9), interleukin-6 (IL-6), and signal transducer and activator of transcription-3 (STAT3) expression levels on MD Anderson metastasis breast-231 cancer (MDA-MB-231) cell lines. Results of an antitumor assay in an in ovo model showed up to 30% inhibition of tumor growth for breast cancer (MDA-MB-231) when using (**5**) (0.650 mg/kg dose) and 17.29% inhibition with the free homoleptic metal complex (1.5 mg/kg dose, (**4**)). While the formulation of inclusion complexes from metal complexes of curcuminoids demonstrates its usefulness in improving the solubility and bioavailability of these metallodrugs, the new compound (**5**) exhibits excellent potential for use as a therapeutic agent in the battle against breast cancer.

## 1. Introduction

Breast cancer is a neoplastic disease with high global mortality for the female population. It is characterized by the accumulation of mutations in the deoxyribonucleic acid (DNA) structure that leads to uncontrolled development and multiplication of cells within the breast tissue [1,2]. This disease will continue to be the cause of many deaths due to tumor recurrence and drug resistance [3]. Therefore, there is a need for the development and formulation of new therapeutic agents to address this type of human ailment.

Curcumin (diferuloylmethane. Figure 1) is one of the significant metabolites present in the rhizome of the Asian spice *Curcuma longa* [4,5]. It has extensive therapeutic properties that are well documented in the scientific area because of its anti-inflammatory [6,7], antioxidant [8], and cytotoxic potential [9]. Therefore, extensive biological studies continue to be a significant task for scientists. 

Recent studies have found that the curcumin molecule is considerably active against breast cancer cell lines (e.g., MCF-7 or MDA) [10,11] and shows antitumoral activity in the ovo model in nanoformulations without altering embryo development [12]. However, disadvantages such as poor solubility [13], low bioavailability [14,15], and rapid metabolism limit its use in the clinic as a successful therapeutic molecule. Consequently, research has focused on curcumin and the synthetic derivatives diacetylcurcumin and dimethoxycurcumin, promoting a considerable increase in new molecules called curcuminoids [8,16,17] or analogs, as shown in Figure 1.

The synthetic curcuminoid obtained from the replacement of two phenolic groups (Ph-OH) with two methoxyl groups (Ph-OCH_3_) is called dimethoxycurcumin (DiMeOC, Figure 1) [18] and has exhibited good cytotoxic activity against cancer cells [19]. In addition, DiMeOC has better metabolic stability [20] (compared with curcumin). Dimethoxycurcumin also exhibits the well-known keto-enol equilibrium at molecular half moiety, which allows it to form chelates or complexes with different metal ions. So, this chemical property has been advantageous in improving its solubility and bioavailability. Different metals of biological interest, such as copper (Cu), iron (Fe), manganese (Mn), zinc (Zn), gallium (Ga), and indium (In), have previously been bonded to curcuminoid-type molecules, forming active metal complexes with antioxidant or cytotoxic activities against breast cancer cell lines (Table 1) [21,22,23,24]. A metal ion related to breast cancer is magnesium (Mg) [25], the second most relevant ion in the human body [26] because it acts as a cofactor for about 300 enzymatic processes involved in maintaining energy metabolism [27], protein synthesis, and DNA replication [28], besides exerting essential antioxidant functions [29].

The ailment hypomagnesemia [30] is related to high oxidative stress and low serum concentrations of magnesium in women with breast cancer that compromises the expression and function of antioxidant enzymes, concomitantly manifesting the progression and proliferation of breast tumors [27,28]. So, considering the importance and participation of magnesium in breast cancer, the synthesis of metallodrugs that contain this metal ion is essential. Surprisingly, only a few scientific reports are found concerning studies of curcuminoid metal complexes against breast cancer cells, as shown in Table 1.

**Table 1 ijms-25-04380-t001:** Curcuminoids and metal complexes against breast cancer.

Compound	Breast Cancer (Cell Line)	Reference
Curcumin	T47D, MDA-MB-415, SK-BR-3, BT-20, MDA-MB-231, MDA-MB-468, MDA-MB-453, MCF-7, MCF-10A	Latifah et al. (2006) [31] Hu et al. (2018) [32] Farghadani et al. (2021) [1] Abdelkader et at. (2022) [33] Ismail-Alhasawi (2022) [34] Fawzy et al. (2024) [35]
DAC	MCF-7	Meza-Morales et al. (2023) [24]
DiMeOC	MCF-10A, MDA-MB-231, MDA-MB 435S, MCF-7	Fuchs et al. (2009) [36] Yoon et al. (2014) [37] Kunwar et al. (2011) [38] Muhammad et al. (2021) [18]
DAC–Magnesium	MCF-7	Meza-Morales et al. (2019) [22]
DAC–Zinc	MCF-7	Meza-Morales et al. (2019) [22]
DAC–Manganese	MCF-7	Meza-Morales et al. (2019) [22]
DiMeOC–Zinc	MCF-7	Meza-Morales et al. (2023) [21]
DiMeOC–Copper	MCF-7	Meza-Morales et al. (2019) [21]

On the other hand, an additional approach to solving the problem of poor solubility of curcuminoid-type molecules has been the discovery of new formulations, including liposomes [39,40], nanoparticles [41,42,43,44,45], or the preparation of inclusion complexes with cyclodextrins [46,47]. Beta-cyclodextrin (BCD) is the first choice [48] for the administration of drugs toward their sites of action because of its low toxicity [49], certain hydrophilicity [50], and adequate cavity. Beta-cyclodextrins can also help to increase the solubility [51] and bioavailability and confer stability [52] of different insoluble guests in aqueous media. 

In this work, we designed a formulation following three concepts of molecular architecture including (a) the use of a metabolically stable curcuminoid compound (dimethoxy curcumin, DiMeOC, (**3**)) and cytotoxic against cancer cell lines; (b) using a homoleptic metal complex with a metal (magnesium, DiMeOC-Mg, (**4**)) of physiological importance related to cancer; and (c) the preparation of an inclusion complex (DiMeOC-Mg-BCD, (**5**)) with beta-cyclodextrin that has adequate cavity size for the encapsulation of a wide variety of drugs. Although the homoleptic metal complexes of curcuminoids have demonstrated high antioxidant and cytotoxic activity in vitro [53], there are few studies of new formulations with cyclodextrins and their biological activity in vivo. 

The overall biological screening suggested focusing on the new inclusion metal complex (**5**) and the free homoleptic metal complex (**4**) using the antitumoral model in ovo. Such a model is a good choice as an alternative for mammalian tumor induction to investigate the characteristics of tumor growth, metastasis, and angiogenesis [54]. The antitumor in ovo assay was carried out by INOVOTION SAS (France).

## 2. Results

Dimethoxycurcumin (**3**) was synthesized according to the “click and unclick chemistry” approach for the synthesis of curcuminoids [55], where the following simple three-step method was used: (i) Synthesis of synthon (**1**) with 2,4-pentanedione and BF_3_**·**THF, which avoids the Knoevenagel reaction by activating the methyl groups. (ii) Condensation of 2 moles of 3,4-dimethoxybenzaldehyde with synthon (**1**). Tributyl borate is used to remove water molecules, while n-butylamine works as a suitable basic catalyzer (see Figure 1). (iii) Removal of BF_2_ with alumina to recover the β-keto-enol function of the curcuminoid (**3**). The synthetic method was adequate for the synthesis of ligands in good (Dimethoxycurcumin-BF_2_ (**2**), 85%) to excellent (DiMeOC (**3**), 90%) yields, as reported previously [24,55]. The spectral information of the precursor (**2**) and ligands (**3**) were confirmed by infrared (IR), mass spectrometry (MS), and nuclear magnetic resonance (NMR). See Appendix A.

The metal complex of dimethoxycurcumin with magnesium (DiMeOC-Mg) was synthesized by previously reported standard methods [22], using magnesium acetate as a source of the metal center and a mixture of ethyl acetate/methanol (7:3) as the solvent (see Figure 2). The stoichiometric relationship between the complex of dimethoxycurcumin and magnesium (**4**) was established by spectroscopic analysis (see Appendix A). NMR contains the signal for two hydrogens of the central methine and four hydrogens of the α,β-unsaturated system; the mass spectra with a peak of 815 *m*/*z* (see Appendix A) that corresponds appropriately to the 1:2 (Mg: (DiMeOC)_2_ (**4**)) homoleptic metal complex. This synthetic approach was suitable for the obtention of the metal complex since a yield of ca. 80% was obtained with purity greater than 98% by high performance liquid chromatography (HPLC). The resulting product from the treatment with beta-cyclodextrin of the metal complex (see Figure 2) was obtained by coprecipitation methodology using acetone and water as solvents. The inclusion complex obtained (DiMeOC-Mg-BCD), stoichiometric ratio, solubility, and purity were established by spectroscopic methods (IR, MS, NMR, diffusion ordered spectroscopy (DOSY), ultraviolet-visible spectroscopy (UV-Vis), HPLC, see Appendix A) and physicochemical analysis in the solid state by ^13^C cross-polarized magic angle spinning (CPMAS)-NMR, scanning electron microscopy (SEM), thermogravimetric analysis (TGA), differential scanning calorimetry (DSC). While the stoichiometry is adequately supported as a 1:1 complex, the exact nature of the spatial molecular association requires further work as it would occur after obtaining a single crystal structure determination.

MDA-MB-231 breast cancer cells were treated with DiMeOC-Mg-BCD. Our findings revealed that this compound showed significant effects in reducing cell survival, affecting cell morphology, and inducing apoptosis. In addition, we examined its effect on cell migration of these cancer cells, showing that this compound suppressed in vitro the cell migration ability in a significant manner, which could confirm the anti-metastatic activity of DiMeOC-Mg-BCD against breast cancer cells [56].

The evaluation of the antitumoral effects of DiMeOC-Mg and DiMeOC-Mg-BCD with cytotoxic activity against the MDA-MB-231 cell lines was performed using the in ovo model. DiMeOC-Mg (1.5 mg/kg) and DiMeOC-Mg-BCD (0.650 mg/kg) reduced the relative tumor weight by 17.29% and 30%, respectively, compared with the control group. 

## 3. Discussion

### 3.1. Infrarred

Characteristic bands for the ligand (DiMeOC (**3**)). One band is observed at 1620 cm^−1^ corresponding to the aromatic ring and another with high intensity at 1506 cm^−1^ for the free carbonyl group of the β-diketone, reveal that the compound exists in enol form, while the band -CH=C- (trans double bond) appears at 967 cm^−1^. The IR spectrum of the magnesium metal complex (DiMeOC-Mg) shows one band at 1627 cm^−1^ assigned to the aromatic rings, one intense band at 1507 cm^−1^ due to the interaction between the metal and the di-carbonyl groups, and an additional band at 470 cm^−1^ assigned to the metal–oxygen bond (M-O) interaction [22]. Beta-cyclodextrin (BCD) showed characteristic bands for the hydroxyl group (-OH) at 3300 cm^−1^, a band for the C-H bond at 2924 cm^−1^, and bands at 1152 cm^−1^ and 1021 cm^−1^ corresponding to C-O bonds of the glucose units as shown in Appendix A [48]. The inclusion complex between DiMeOC-Mg and BCD shows shifts in the wave numbers of the bands related to BCD, i.e., from 3300 cm^−1^ to 3386 cm^−1^, from 2924 cm^−1^ to 2930 cm^−1^, from 1152 cm^−1^ to 1157 cm^−1^, and from 1021 cm^−1^ to 1024 cm^−1^. The band related to the magnesium metal complex is at 1510 cm^−1^. These spectroscopic changes [57] confirm the formation of the new complex (DiMeOC-Mg-BCD) involving the interaction between the homoleptic magnesium metal complex (guest) and the BCD (host).

### 3.2. Mass Spectrometry

The mass spectrometry confirmed the molecular weight of all compounds, revealing the *m*/*z* = 444 ratio for DiMeOC-BF_2_, corresponding to the difluoro-borated precursor C_23_H_23_BF_2_O_6_. The ligand DiMeOC shows a *m*/*z* = 397 ratio corresponding to the molecular formula C_23_H_24_O_6_. The magnesium metal complex (DiMeOC-Mg) has a *m*/*z* = 817 ratio corresponding to the molecular formula C_46_H_46_MgO_12_, while the inclusion complex exhibits a *m*/*z* = 1949 ratio corresponding to an inclusion complex with the molecular formula of C_88_H_116_MgO_47_ and a stoichiometric ratio of DiMeOC-Mg: BCD 1:1.

### 3.3. ^1^H NMR Liquid State

The synthetic difluoroborated precursor (DiMeOC-BF_2_) obtained by the condensation of two moles of 3,4-dimethoxybenzaldehyde with synthon 1 was verified by the formation of the heptanoid chain. The chemical shifts reveal two double bonds with d = 7.12 ppm (α) and d = 7.97 ppm (β), *J* = 16 Hz, corresponding to *trans* vinyl protons. The central methine proton (singlet) is found at d 6.50 ppm and the methoxyl groups appear at d = 3.84 ppm. 

The curcuminoid ligand (DiMeOC) presented the characteristic signals of a 1,3-dicarbonyl α,β-unsaturated system, and the protons d = 6.84 ppm (α) and d = 7.59 ppm (β) were observed at lower frequencies with respect to the precursor due to the elimination of the BF_2_ group. In addition, the methine proton was observed at d = 6.11 ppm and the enolized proton (-OH) at d = 16.33 ppm (characteristic for most curcuminoids), revealing the strong hydrogen bond between the β carbonyls.

The homoleptic metal complex was confirmed by ^1^H NMR in the liquid state. In general, protons of metal complexes (curcuminoid type) have a chemical shift at lower frequencies with respect to the ligand, attributed to an increase in electron density from the metal (magnesium) in the complex. Vinylic hydrogens (d = 6.71 ppm and d = 7.34 ppm) appear at 50 and 100 Hz lower frequencies, respectively, and the central methine proton (d = 5.63 ppm) appears significantly shifted to lower frequencies (ca. 200 Hz) with respect to the ligand.

The changes in chemical shifts in the free BCD and its complex form were determined by ^1^H NMR. The liquid NMR spectra were obtained in dimethylsufoxide-*d*_6_ (DMSO-*d*_6_, see Appendix A) and deuterium oxide (D_2_O). ^1^H NMR spectra in DMSO-*d*_6_ for free BCD and BCD in the inclusion complex showed signals in agreement with the expected compounds [58]. However, substantial differences in chemical shifts for hydrogens related to the BCD’s cavity were not observed, which is consistent with previous reports [59]. 

The work of Kida (2011) et al. [60] describes that a guest (pyrene) may have different inclusion modes with cyclodextrins using different deuterated solvents (benzene or cyclohexane). It is essential to highlight that when using benzene-*d*_6_, the guest associates in the secondary region of cyclodextrin through π-π interactions may form a “sandwich”-type complex. Still, with cyclohexane-*d*_12_, the guest resides in the cavity of cyclodextrin, thus forming a different type of inclusion complex. From these observations, it is clear that the solvent significantly affects the mode of inclusion between guest and host molecules. Therefore, differences in chemical shifts for protons in the host cavity, using one solvent or another, are expected.

Based on the above, we made comparisons of the chemical shifts of the inclusion complex in deuterium oxide (D_2_O), obtaining the ^1^H NMR spectra shown in Figure 2.

The chemical shifts in BCD in the complex form are affected and shifted to lower frequencies due to the protection conferred by the aromatic rings of the guest. Furthermore, when there is an interaction between guest and host, the chemical shifts in H3 (−0.06 ppm) and H5 (−0.08 ppm) are expected to change, as shown in Table 2, and correlate with a total inclusion complex. These results support the successful formation of the inclusion complex (DiMeOC-Mg-BCD) and are confirmed by other spectroscopic techniques, as shown below. 

### 3.4. DOSY Spectra

The different diffusion values of the molecular species in the DMSO-*d*_6_ spectrum follow the formation of the inclusion complex. In the DOSY experiment, the diffusion becomes smaller when the species possesses a greater hydrodynamic radius (see Table 3). The diffusion coefficient for the BCD inclusion complex has the lowest value (8.34 × 10^−11^ cm^2^/s), while the free form has 1.26 × 10^−10^ cm^2^/s, and it agrees well with reported values [58].

The DOSY spectrum of DiMeOC-Mg-BCD (Figure 3b)) reveals the association–dissociation effects with differentiated signals (see diffusion line). Because interactions in the inclusion complex are of a non-covalent type (e.g., hydrogen bonds, VdW forces) [61], this reflects the equilibrium in solution between the host (BCD) and guest (DiMeOC-Mg). 

### 3.5. Solubility, Ratio of Guest–Host, and HPLC

The increase in solubility in aqueous media is one of the most relevant points of inclusion complexes due to their multiple applications in the medical area. The solubility of DiMeOC-Mg-BCD (expressed in µg/mL) was determined using a standard curve and measuring absorbance at 415 nm (see Appendix A) in the aqueous cell culture medium (PBS) and compared with DiMeOC-Mg. A 6-fold increase in solubility of the metal complex was found in the inclusion complex (see Table 4). Solubility studies have shown that curcumin-based inclusion complexes form a uniform dispersion in aqueous media because of the good compatibility between cyclodextrin and curcuminoids [62].

The inclusion complex corresponds appropriately to a 1:1 ratio (guest:host) of the magnesium metal complex (**4**) with beta-cyclodextrin and can be verified when a maximum molar concentration (415 nm, UV–Vis) of the compound is obtained with the mole fraction corresponding to 0.5 of BCD (see Appendix A) [63].

The HPLC chromatograms (see Appendix A) show that DiMeOC-Mg-BCD’s retention times in the acetonitrile/water elution medium have decreased (19 min) compared to DiMeOC-Mg’s (20 min), demonstrating that the new inclusion complex’s solubility in aqueous media has improved. Furthermore, the association between the guest and BCD is strong since only one single component of high purity is present in the chromatogram. 

### 3.6. ^13^C CPMAS Solid-State NMR

The structural differences among the synthesized complexes were studied using ^13^C CPMAS ssNMR analysis, showing differences in shape and signal intensity [64]. The most significant change between the ligand (DiMeOC) and its magnesium metal complex (DiMeOC-Mg) resides in the region of the carbonyl groups; for DiMeOC (ligand), two different carbonyl groups are observed at 185 ppm and 189 ppm (see Figure 4). However, DiMeOC-Mg shows the carbonyl signals as a singlet at 187 ppm due to the coordination with magnesium. Furthermore, this can be associated with increased symmetry at the coordination plane around the metal atom [24].

Furthermore, to evaluate the spectral differences in the inclusion complex, we performed a comparison of three ^13^C CPMAS ssNMR spectra (DiMeOC-Mg, BCD, and DiMeOC-Mg-BCD). The inclusion complex (Figure 4, number 4) shows a very low signal intensity in the zone corresponding to the guest molecule, while the signals that correspond to the host (BCD) are clearly observed with higher intensity and multiplicity. A broadening effect of the signals corresponding to the guest is observed, especially for the carbonyl groups (186 ppm) as well as the region corresponding to the aromatic and vinyl groups (110 ppm–160 ppm). The signals between 140 ppm and 150 ppm that correspond to the guest practically disappear.

The above analysis of spectra ^13^C CPMAS NMR may lead to the following three important conclusions:The guest (DiMeOC-Mg) was adequately complexed with BCD using the selected coprecipitation method.The relaxation phenomena for the guest molecule becomes very fast (low intensity of signals) when located at the cavity of BCD.The inclusion complex reveals an amorphization (broadening of signals) in agreement with the data obtained by scanning electron microscopy (SEM).

### 3.7. SEM Analysis

Scanning electron microscopy (SEM) was used to characterize the visual and morphological changes from the starting materials to the inclusion complex. Both beta-cyclodextrin and the ligand (DiMeOC) show crystalline morphology in the form of bars or elongated bars. However, the magnesium complex (DiMeOC-Mg) appears as an amorphous and irregular material (Figure 5c). 

The SEM analysis of DiMeOC-Mg-BCD (Figure 5d), shows morphological changes that point out the formation of the inclusion complex due to the interaction between BCD and the magnesium metal complex [62]. The new inclusion complex exhibits a significant change in morphology and consists of amorphous material where regular crystal habits are absent.

### 3.8. TGA and DSC Analysis

The thermal analysis results allowed for evaluating the differences in and stability of the starting materials compared to the inclusion complex (DiMeOC-Mg-BCD). Thus, TGA and DSC showed differences between the starting materials (BCD and DiMeOC-Mg) and the inclusion complex.

BCD presents three stages of weight loss (Figure 6). The first loss (−13%) is due to the loss of water molecules in a temperature range of 25–150 °C, the second loss of 64% (300–360 °C) corresponds to the solid transition to the liquid and a partial decomposition, and for the final stage (>380 °C), there is a minimum weight loss (−8.3%) due to the total thermal decomposition of the BCD. The magnesium metal complex presents an initial weight loss of 5% (25–150 °C) due to the evaporation of water. However, its melting point and decomposition partially occur between 230 and 420 °C with a weight loss of 34%, and above 440 °C, the thermal decomposition of the compound occurs.

Inclusion complexes are frequently amorphous solids with reduced crystallinity where high hydration levels can be associated. DiMeOC-Mg-BCD initially loses 8% of weight (water molecules of hydration, 25–160 °C) and agrees well with a successful host–guest inclusion. We observed that at the initial stage, a 10% weight loss occurs at a higher temperature range (near 160 °C, Figure 6); this suggests a good encapsulation efficiency [52,62]. Furthermore, the 47% partial weight loss (280–360 °C) distinguishes it from the initial starting materials, resulting in a more stable product than BCD for the same temperature range. Additionally, the DSC thermogram (Figure 7) shows that BCD has a prominent endothermic peak at 325 °C, while the homoleptic metal complex shows it at 248 °C, revealing a phase transition. The new inclusion complex (DiMeOC-Mg-BCD) has a prominent endothermic peak at 334 °C, and the peak at 248 °C in the new thermogram disappears and can be attributed to the guest being wholly included in the BCD with the replacement of water molecules [65]. In addition, the change in the profile of the thermogram, which includes new phase transitions, reveals the specific profile of the new DiMeOC-Mg-BCD inclusion complex.

The apices of melting points for DiMeOC-Mg and DiMeOC-Mg-BCD are found at 248 °C and 334 °C respectively (Figure 7). However, DiMeOC-Mg shows a smaller fusion area (21.46 J/g) compared with DiMeOC-Mg-BCD (257.6 J/g), indicating that the endotherm of the new inclusion complex belongs to an amorphous solid profile with a wide melting point range.

### 3.9. Cytotoxic Activity in Human Tumour Cells and Evaluation of Morphological Changes

We carried out cytotoxicity studies on MDA-MB-231 triple-negative breast cancer cells because these cells are commonly used to model late-stage breast cancer and are characterized by being invasive and spontaneously generating metastasis in lymph nodes [66]. DiMeOC alone did not show significant cytotoxic activity. However, DiMeOC-Mg showed increased cytotoxic activity towards these breast cancer cell lines. We attribute the increase in cytotoxic effect to the slightly higher metal complex solubility in aqueous media (Table 5).

Furthermore, recent studies by Yin-Yin et al. [3] on the inhibitory effects of an analogous curcuminoid (EF-24) in MDA-MB-231 cell lines demonstrated that cell death is initiated by crosstalk between autophagy and apoptosis, with induction of mitochondrial apoptosis. This observation led us to suggest that the value of 50% cytotoxic concentration (CC_50_ = 10.73 ± 0.1) found for the formulated inclusion complex (DiMeOC-Mg-BCD) may be related to the triggering of non-necrotizing cell death pathways.

Giemsa stain and phase contrast microscopy were applied to evaluate morphological alterations in MDA-MB-231 cells. After 48 h of treatment, the cell lines had unchanged size and morphology in both the control and vehicle. In contrast, MDA-MB-231 cell lines treated with DiMeOC-Mg-BCD showed cell size reduction (shrinkage), nuclear condensation (dark condensed and rounded nuclei), and the presence of membrane-bound apoptotic bodies (shown in Figure 8). Also, a significant reduction in cell number and the presence of nonadherent cells and cellular fragments appeared. The morphological alterations produced by curcumin and its derivates suggest different stages of cell death, such as apoptosis [67], lost cell integrity, promotion of cancer cell progression, and nucleic acid damage, which could be related to a downregulation of transcription factors and gene expression and inhibition of tumor growth and angiogenesis [68].

### 3.10. Esterase Activity and Membrane Damage Induced by DiMeOC-Mg-BCD on MDA-MB-231

To elucidate the action mechanism of DiMeOC-Mg-BCD on MDA-MB-231, the fluorescent probe (Calcein-AM and Eth-1) for esterase activity and cell membrane damage was used to confirm that this compound affects cell viability by flow cytometry [69]. The discrimination of live and dead/necrotic cells by this method considers a decreased activity of esterase, which is an early characteristic in apoptosis, followed by depolarization of mitochondrial membrane potential (ΔΨm), overexpression of caspases enzymes, the symmetrical distribution of phosphatidylserine and lipid rafts on the plasma membrane, and the loss of membrane integrity [70]. DiMeOC-Mg-BCD caused a significant decrease in esterase activity/viable cells (C-AM+) of breast cancer cell lines (63.57%, CC_50_, 48 h). The percentage (56.91%) of membrane damage cells/dead cells (Eth-1+) was statistically significant compared with the DMEM and DMSO controls, but they did not affect cell viability (see Figure 9).

### 3.11. Apoptosis Induced by DiMeOC-Mg-BCD on MDA-231-MB Breast Cancer

MDA-MB-231 cell lines were stained with annexin V-FITC (AV+) and propidium iodide (PI+) and analyzed using flow cytometry to discriminate between apoptotic and necrotic cell death. Annexin V is a protein conjugated to a green fluorescent dye to detect apoptosis. Propidium iodide (PI) is a red fluorescent dye that stains the DNA of both necrotic and late apoptotic cells with damaged membranes (see Figure 10) [32].

Figure 11 shows the following: (a) DMEM and vehicle (1.3% DMSO) giving 70.4% and 82.25% of live cells (AV−/PI−); DiMeOC-Mg-BCD reduces this population to 4.03%; (b) 5% DMSO produces 54.3% of early apoptotic cells (AV+/PI−), and was used as the positive control; (c) DiMeOC-Mg-BCD reached 87.625% of late apoptotic cells (AV+/PI+) in contrast with the DMEM percentage (9.7%) and vehicle (16.16%); and (d) heating at 65 °C was used as the necrotic control, showing 87.625% of AV−/PI+ cells. Our results indicate that early apoptosis is the cell death mechanism induced by DiMeOC-Mg-BCD.

### 3.12. Effect of DiMeOC-Mg-BCD on MDA-MB-231 Cell Migration (Wound-Healing Assay)

Increased cell migration is a critical physiological process in tissue repair, immune responses, and development. In cancer studies, the increase in cell migration is a hallmark of migratory ability, metastasis, and a bad clinical prognosis [2,71,72]. We examined the effect of DiMeOC-Mg-BCD on motility in breast cancer cell lines by a wound-healing assay for the MDA-MB-231 triple-negative cancer cell lines (Figure 12).

We found that this compound significantly inhibited the ability of cells to close the artificial wound in the 48 h treatments, reducing the relative migration ratio concerning the vehicle control and without a significant difference between 24 h and 48 h, demonstrating a reduction over time. The wound size for the vehicle reached up to 42.4% and 66% in the MDA-MB-231 control after 24 h and 48 h, respectively, while DiMeOC-Mg-BCD showed 32.4% after 24 h exposure at 1 µM and 47% after 48 h. Our results show that DiMeOC-Mg-BCD can repress the motility of breast cancer cells.

### 3.13. Effect of DiMeOC-Mg-BCD on MMP-2, MMP-9 and IL-6, and Total STAT3 Gene Expression on MDA-MB-231

The inhibition expression of matrix metalloproteinase as MMP-2 and MMP-9 expression levels in breast cancer tissues correlates with lymph node metastasis, tumor growth, angiogenesis, invasion, and metastasis [56]. We analyzed the expression of MMP-2 and MMP-9 on MDA-MB-231 cancer cells after treatment with DiMeOC-Mg-BCD (at CC_50_) for 48 h by RT-PCR. As shown in Figure 13a,b, MMP2 and MMP9 protein expression significantly decreased (*p* < 0.001) after normalization to the vehicle control, evidencing that this compound effectively inhibited migration and invasion on MDA-MB-231. Curcumin or curcuminoids are beneficial in inhibiting cell proliferation, migration, and metastasis, modulating cancer-related pathways, and sensitizing cells to radiotherapy and chemotherapy in vivo, which could be evaluated for this compound since metalloproteins expression is used as a reference index for guiding breast cancer progression and treatment efficiency [56]. 

Metastasis in breast cancer occurs when cancer cells detach from the primary tumor, invade adjacent tissues to reach the bloodstream, and colonize distant organs [56]. The development of distant metastasis is a crucial event that limits the survival of breast cancer patients. The epithelial–mesenchymal transition (EMT) enables breast cancer cells to exhibit a self-renewal characteristic and to enhance cell motility/migration, which results in the occurrence of metastatic colonies at distant sites [2,71]. 

Many studies documented that STAT-3-mediated breast cancer metastasis happens through the upregulation of MMP9 and MMP2 expression. Targeting MMP9 can reduce breast cancer progression and modulate EMT genes [1,73]. In the present study, DiMeOC-Mg-BCD was found to inhibit the expression of MMP9 and MMP2 genes in the MDA-MB-231 breast cancer cell line.

We evaluated the expression of IL-6 and total STAT3 gene expression in MDA-MB-231 cells treated with DiMeOC-Mg-BCD for 48 h. IL-6 and total STAT3 gene expression decreased significantly compared with the vehicle control cells (see Figure 13c,d). Taken together, the interleukin-6 (IL-6) and the signal transducer and activator of transcription 3 (STAT3) or IL-6/STAT3 pathway is a central regulator of breast cancer metastasis by promoting breast cancer cell proliferation, the development of metastasis (by the epithelial–mesenchymal transition (EMT) and enriching cancer stem cells (CSCs)), and suppressing apoptosis. They are also related to therapeutic resistance [74].

Curcuminoid complexes downregulate STAT3 and IL-6 expression and inhibit STAT3 translocation into the nucleus, suppressing cell proliferation, invasion, and metastasis in hormone triple-negative receptor breast cancer [1]. Additional studies have observed that curcumin inhibits IL-6/JAK/STAT3 signaling by significantly reducing the phosphorylation of JAK2 and STAT3 levels in colorectal cancer (CRC) cells. Curcumin (2 g/day) and chemotherapeutic drugs were synergistically administered in 27 patients with CRC during a safe and tolerable clinical trial. The increased patients’ overall survival and response rate makes DiMeOC-Mg-BCD a promising compound for in vivo preclinical trials [68].

IL-6 expression is a critical step of breast cancer metastasis, leading to the activation of the JAK2/STAT3 signaling pathway that promotes proliferation, invasion, metastasis, and angiogenesis and inhibits apoptosis in breast cancers. Thus, a drug targeting this pathway may benefit breast cancer therapies [1]. So, to check the activity of DiMeOC-Mg-BCD on cancer cell lines at the molecular level, the IL-6/STAT3 signaling pathway was investigated. Our data showed that this compound reduced the expression of IL-6 mRNA and STAT3, which promote proliferation, metastasis, invasion, and angiogenesis in breast cancers, and could result in the suppression of the NF-kB/IL-6/JAK2/STAT3 signaling pathway.

The activated IL-6/STAT3 pathway can inhibit caspase-dependent apoptosis, promoting proliferation and metastasis of cancer cells [74]. Thus, targeting apoptosis became a therapeutic approach in cancer [1,32,74]. DiMeOC-Mg-BCD decreased the expression of metalloproteins, IL-6, and STAT3 in the MDA-MB-231 breast cancer cell line, leading to the induction of apoptosis. Figure 14 summarizes the tentatively proposed action mechanism for the anti-proliferative, EMT inhibition, and apoptotic activity of DiMeOC-Mg-BCD against the MDA-MB-231 triple-negative breast cancer cell line. To confirm the role of this compound, which is suggested to inhibit the IL-6/STAT3 action mechanism with subsequent initiation of apoptosis [75], it would be necessary to carry out further experiments such as a Western blot study to detect the participation of proteins involved in the proposed signaling pathway.

### 3.14. Antitumoral Activity and Embryotoxicity

After ten days of implantation of tumor cells on the Chorioallantoic membrane (CAM), tumors became detectable. Treatments corresponding to different groups were administrated as described in Table 6. The increase in the inclusion complex’s solubility (in aqueous media) correlates appropriately with an increase in bioavailability, observing a greater antitumor effect of the new inclusion complex (DiMeOC-Mg-BCD, 30%, see Figure 15).

In our work, designing and synthesizing a homoleptic metal complex of dimethoxycurcumin with magnesium (DiMeOC-Mg) and their inclusion complex (DiMeOC-Mg-BCD) demonstrated a promising strategy for improving its cytotoxic effect and partial tumor regression in ovo (see Figure 15). Curcumin and curcuminoids are of broad interest as cytotoxic agents against several cancer cell lines or as antitumoral compounds. Research about new curcuminoid formulations aims to increase their bioavailability. 

DiMeOC-Mg and DiMeOC-Mg-BCD did not produce any abnormality or embryo death. Our embryotoxicity results agree with Strojny et al. [12], who reported that curcumin did not produce any sign of embryotoxicity. Figure 16 shows that dead embryos were present in the control, paclitaxel, DiMeOC-Mg, and DiMeOC-Mg-BCD groups at different doses unrelated to experimental compounds (*p* = 0.6540). No abnormality was observed in chicken embryos.

## 4. Materials and Methods

All the chemicals were commercially available from Sigma-Aldrich (Toluca, México) and were used without a purification process. Melting points were recorded using an electrothermal engineering IA9100X1 melting point apparatus, and the values were uncorrected [76].

The spectra IR-ATR determinations were recorded in the 4000–400 cm^−1^ range using FT-IR NICOLET IS-50 Thermo Fisher Scientific (Waltham, MA, USA) spectrophotometer equipment.

Mass spectra were obtained using a JEOL SX 102 A spectrometer (JEOL de México S.A. de C.V., Mexico City, Mexico) equipped with MALDI-Flight time technology or using MStation JMS-700 JEOL equipment (Electron Ionization, 70 eV, 250 °C, impact positive mode and calibration standard with perfluorokerosene) and AccuTOF JMS-T100LC JEOL equipment (DART^+^, 350 °C, positive ion mode and calibration standard with PEG 600) [21,77].

^1^H and ^13^C NMR liquid spectra were acquired in dimethyl sulfoxide (DMSO-*d*_6_) or deuterium oxide (D_2_O) on a Bruker Fourier 400 MHz (Billerica, MA, USA) spectrometer with TMS serving as the internal reference, and NMR spectra were processed with MestreNova software 12.0.3.3. [78]. 

DOSY was acquired with JEOL 400 MHz (JNM JEOL ECZ 400S) equipment (JEOL de México S.A. de C.V., Mexico City, Mexico), using the pulse sequence One-shot-DOSY, with grad_1_amp 30 (min) and grad_2_amp 280 (min), with the following parameters: sw = 20 ppm; at = 3.0 s; d1 = 5.0 s; nt = 32; lb = 0.1 Hz; diffusion time 500 ms. For the NMR data acquisition, Jeol Delta 6.0 software was used, and for data processing, MestreNova software 12.0.3.3. (Mestrelab Research) was used [78].

The maxima absorption measurements were recorded with a UV–Visible Shimadzu, U160 spectrophotometer. The aqueous (PBS) solubility of DiMeOC-Mg and DiMeOC-Mg-BCD were determined by the shake-flask method in triplicate (see Appendix A). The quantification was performed by interpolation using a previously constructed calibration curve (see Appendix A), and the determinations were analyzed by UV–Vis spectroscopy at 415 nm with enzyme-linked immunosorbent assay (ELISA) plate reader equipment (Bio-Tek Instruments, Winooski, VT, USA). Phase solubility studies were carried out according to the Higuchi and Connors methods [79] (see Appendix A). Briefly, 2 mg of DiMeOC-Mg was added to a series of different mole fractions of BCD (0.0, 0.125, 0.25, 0.5, 0.75, 1.0, and 1.25) in PBS medium. After being shaken for 48 h, the solution was filtered through a sintered filter of 0.45 μM, and the amount dissolved in each solution was measured at 415 nm. 

HPLC chromatograms were recorded using an Agilent 1260 infinity II with diode -UV detector at 417 nm, column Spherisorb 25 cm × 4.6 mm × 5 μm particle size, eluting with a solvent isocratic acetonitrile/water (formic acid 0.02%) 55/45 and flow 1.0 mL/min [80].

^13^C CPMAS ssNMR spectra were recorded using a JEOL 600 MHz spectrometer (15.0 kHz of MAS) with adamantane as the reference (298 K). 

Surface morphology images were recorded for BCD, DiMeOC, DiMeOC-Mg, and DiMeOC-Mg-BCD, and for the Scanning Electronic Microscopy (SEM) analysis, the images were captured with a VEGA3 model microscope (TESCAN, Brno, Czech Republic). 

TGA and DSC analysis was carried out with a thermobalance (thermo analyzer Netzsch model STA 449 F3 Jupiter) using an aluminum crucible 25/40 µL, outer bottom Ø 5 mm (NETZSCH). The sample was heated from 25 °C to 550 °C at a heating rate of 10 °C min^−1^ under nitrogen atmosphere [62].

The human breast epithelial cell line MDA-MB-231, an estrogen receptor-negative cell line derived from a metastatic carcinoma obtained from the American Type Culture Collection (ATCC; Manassas, VA, USA), was used in this study. MDA-MB-231 cells in Dulbecco’s Modified Eagle Medium (DMEM) medium were supplemented with 10% fetal bovine serum (FBS), 2 mM L-glutamine, 10,000 units/mL penicillin G sodium, 10,000 μg/mL streptomycin sulfate, 25 μg Amphotericin B (Gibco, Waltham, MA, USA), and 1% of non-essential amino acids (Gibco, Waltham, MA, USA). The cell lines were kept at 37 °C in a humidified atmosphere of 5% CO_2_, and cell viability exceeded 95%.

We used a protein-binding dye sulforhodamine B (SRB) colorimetric assay [81]. A suspension of 100 μL containing 5000–10,000 cells per well was cultured into 96 micro-litre plates (Costar). Different concentrations of DiMeOC, DiMeOC-Mg, DiMeOC-Mg-BCD, and control vehicle DMSO 1.3%, incubated at 37 °C for 48 h in a 5% CO_2_ atmosphere, were used. Subsequently, the cells were fixed on the plastic substratum by adding 50 μL of cold aqueous 50% trichloroacetic acid for 48 h. Then, the cells were removed from the tissue culture flask by treatment with trypsin and diluted with fresh media from the cell. The plates were incubated at 4 °C for one hour, washed with water, and air-dried. The addition of 4 % SRB stained the trichloroacetic acid-fixed cells. The free SRB was removed by washing with 1% aqueous acetic acid, the plates were air-dried, and the dye was solubilized by adding 10 mM unbuffered Tris-base (100 μL). The plates were placed on a shaker for 10 min, and the absorption was measured at 515 nm using an ELISA plate reader (Bio Tex Instruments, Winooski, VT, USA). The cell viability of MDA-MB-231, cells cultured in the presence of the assessed compounds was calculated as a percentage of the control cells, and the CC_50_ values were obtained from dose–response curves. All experiments were performed in triplicate, and CC_50_ was calculated using GraphPad Software 6.0 (GraphPad Inc., San Diego, CA, USA). The results are expressed as the mean of CC_50_ relative to vehicle and control.

To evaluate the morphological alterations induced by the DiMeOC-Mg-BCD complex on MDA-MB-231, cell lines were cultured on a tissue culture flask with DiMeOC-Mg-BCD (CC_50_) and different treatments for 48 h. After incubation, the cells were washed two times with PBS, fixed with methanol 100%, and stained with Wright-–Giemsa (Sigma-Aldrich, WG32-1L, Darmstadt, Germany). The culture flasks were examined under an Olympus IX71 Inverted Fluorescence Phase Contrast Microscope. The images were recorded with a Nikon Coolpix 4300 digital camera using 20x objectives and analyzed by ImageJ 1.52a software [82].

Esterase activity and membrane damage were determined using a LIVE/DEAD Viability/Cytotoxicity Kit (Molecular et al., Eugene, OR, USA). MDA-MB-231 (0.6 106/mL) was culture and incubated at 37 °C in 24 well plates and treated with DiMeOC-Mg-BCD (CC_50_). After 48 h of incubation, the cells were centrifuged at 3000 rpm for 10 min to remove the supernatant medium; the pellet was next resuspended in 997 µL of cold phosphate-buffered saline (PBS), 2 µL of a solution of Calcein-AM (C-AM) 50 µM, and 1 µL of ethidium homodimer (Eth-1) 2 mM. Then, the samples were incubated for 45 min at room temperature and immediately analyzed in a ThermoFisher Attune Flow Cytometer with a 530/30 nm filter (FL1-H) for calcein (green fluorescence/living cells) and a 670 nm/long-pass filter (FL3-H) for Eth-1 (red fluorescence/dead cells); 20,000 events per treatment were acquired. The data were analyzed using FlowJo 7.3.2 software [83] and expressed as the percentage of cells for each population phenotype. The compensation was performed using live cells grown in DMSO, unstained and stained with C-AM, and dead cells by heat (65 °C for 15 min) stained with Eth-1 as a dead cell control.

Double staining for annexin V-fluorescein isothiocyanate (AV-FITC) and propidium iodide (PI) was performed with an annexin-V apoptosis detection kit (Molecular Probes, Eugene, OR, USA). MDA-MB-231 cells were treated with DiMeOC-Mg-BCD (CC_50_) or control and vehicle DMSO for 48 h. The cells were washed twice in cold annexin V-buffer and centrifuged at 3000 rpm for 10 min. The pellets were resuspended in 20 μL of annexin V FITC, and after 15 min of incubation in the dark, 480 μL of annexin V-buffer containing 0.5 mg/mL of PI was added according to the manufacturer’s instructions. Annexin V-FITC labeling was recorded on a ThermoFisher Attune Flow Cytometer and analyzed using FlowJo 7.3.2 software [83] (Rodríguez-Hernández et al., 2020).

To evaluate the effect of DiMeOC-Mg-BCD on cell migration, a wound-healing assay was carried out in a mammary cancer-derived cell line (MDA-MB-231). A total of 2 × 105 cells per well were seeded in a 24-well plate and cultured at 38 °C and 5% CO_2_ until confluence reached 90–100%. Then, an artificial space called a wound was made on the cell monolayer using a 200 μL sterile micropipette tip, which was then washed with PBS to remove suspended cells, and the supernatant was removed and replaced with a new supplemented medium with DiMeOC-Mg-BCD and vehicle dimethyl sulfoxide (DMSO) and incubated for 48 h. We tested with DiMeOC-Mg-BCD (at 1 μM). The images were captured by an inverted microscope (DIAPHOT 300 Nikon^®^, Tokyo, Japan) with a digital camera (AmScope MD500, Irvine, CA, USA) at 0 h, 24 h, and 48 h of treatment. Wound areas were obtained using polygon selection and the measure tool of ImageJ 1.52a software [82]. The relative migration ratio (%) (RMR) = ((Wound area 0 h − Wound area 24 h or 48 h) ÷ (Wound area 0 h) × 100). The results express triplicated experiments, and significance was obtained by a two-way ANOVA with Tukey’s multiple comparisons test.

For the gene expression study, the cells were plated at a density of 150,000 cells in six-well plates with DMEM. Then, the cells were incubated with DiMeOC-Mg-BCD for 48 h, and RNA was extracted for gene expression studies. Gene expression was studied by extracting total RNA from the treated cells using Trizol [84]. In all cases, the amount and quality of RNA were estimated spectrophotometrically at 260/280 nm using a Synergy HT (Biotek, Shoreline, WA, USA), and a constant amount of RNA (2 µg) was reverse transcribed using the Maxima First Strand cDNA Synthesis kit for RT-qPCR (Thermo Scientific™, LT, Waltham, MA, USA) according to the manufacturer’s instructions. Primers and probes for qPCR amplifications were designed by the Universal Probe Library Assay Design Center from Roche, and the respective sequences are listed in Table 7. Identical RT-qPCR conditions were performed for all genes, and in all cases, the results were normalized against ribosomal protein L32 (*RPL32*) used as a housekeeping gene. Real-time RT-qPCR amplifications were carried on a LightCycler^®^ 480 II (Roche, Basel, Switzerland), as previously described [85]. 

The statistical analyses were performed using Prism 6 software (GraphPad, San Diego, CA, USA) [86]. All the experiments represent the mean of three independent assays tested in duplicate, and the data shown in the graphs are expressed as the mean ± standard deviation (SD). The data were analyzed using one-way analyses of variance (ANOVAs). Significant differences among means were identified using Dunnett’s multiple comparisons tests. Values of *p* < 0.001 and *p* < 0.05 were considered statistically significant. For gene expression, the results are expressed as the mean ± S.D. Statistical differences were determined by one-way ANOVA followed by appropriate post hoc tests (Holm–Sidak method for pairwise comparisons) using a specialized software package [87] (SigmaPlot 11.0, Jandel Scientific, Palo Alto, CA, USA). Experiments were performed from three separate cell cultures, and each variable was assessed in triplicate. Differences were considered statistically significant at * *p* < 0.05.

Antitumor tests were carried out in the in ovo model by the company INOVOTION SAS, 5 Avenue du Grand Sablon 38700 La Tronche, France. Results were delivered with the following numbers of study: STU20220110, STU20220112, and STU20221017_CU. Double-blind testing was used, labeling the chemical compounds as FOIN-REH-M3 (Phase 1, corresponding to DiMeOC-Mg) and FOINS-REH-M3 (Phase 2, corresponding to DiMeOC-Mg-BCD). The methods used are briefly described here.

In ovo chicken embryo experiment. Fertilized white Leghorn eggs were incubated at 37.5 °C with 50% relative humidity for nine days. At that moment, the CAM was dropped by drilling a small hole through the eggshell into the air sac, and a 1 cm^2^ window was cut in the eggshell above the CAM. At least 15 eggs were grafted for each group because 10–15% of death may occur by invasive surgical manipulation.

Amplification and grafting of tumor cells. The MDA-MB-231 tumor cell line was cultivated in a DMEM medium supplemented with 10% FBS and 1% penicillin/streptomycin. On day 9, the cells were detached with trypsin, washed with a complete medium, and suspended in a graft medium. An inoculum of 1 × 10^6^ cells was added onto the CAM of each egg, and then eggs were randomized into groups.

Quantitative evaluation of tumor growth. On day 18, the tumor was carefully removed, washed with PBS buffer, and then directly transferred to a 4% paraformaldehyde solution for fixation for 48 h. After that, the tumors were weighed. A one-way ANOVA analysis with Student–Newman–Keuls post-test was performed on the data (<0.05).

Quantitative evaluation of embryonic toxicity. Embryonic viability was checked daily. The number of dead embryos was counted on day 18 in combination with the observation of eventual visible gross abnormalities to evaluate treatment-induced embryotoxicity. Any visible abnormality observed during this study was also briefly described. A Kaplan–Meyer curve was used to evaluate the final death ratio.

### Synthetic Procedures

The synthon was prepared according to a previously reported methodology, and the spectroscopic data agreed satisfactorily with previously reported data [55]. 2,2-difluoro-4,6-dimethyl-2H-1,3,2-dioxaborinin-1-ium-2-uide (Synthon **1**), yield 95%, solid amber, melting point 40 °C. ^1^H NMR (400 MHz, CDCl_3_, TMS): d 5.96 (s, 1H), 2.27 (s, 6H); ^13^C NMR (100 MHz, CDCl_3_, TMS): d 192.63, 102.12, 24.32. These data coincide with the literature [55]. 

The aldolic condensation of 3,4-dimethoxybenzaldehyde (2 moles) with **1** was carried out according to the experimental conditions reported in the literature [88,89].

Dimethoxycurcumin-BF_2_, **2**: 4,6-bis((E)-3,4-dimethoxystyryl)-2,2-difluoro-2H-1λ3,3,2λ4-dioxaborinine: yield 85%, violet powder, m.p. = 226 °C. ^1^H NMR (400 MHz, DMSO-*d*_6_) δ 7.97 (d, *J* = 15.6 Hz, 2H), 7.50 (d, *J* = 2.0 Hz, 2H), 7.47 (dd, *J* = 8.3, 2.0 Hz, 2H), 7.12 (d, *J* = 15.6 Hz, 2H), 7.08 (d, *J* = 8.3 Hz, 2H), 6.50 (s, 1H), 3.84 (s, 12H). ^13^C NMR (100 MHz, DMSO-*d*_6_) δ 179.07, 152.58, 149.13, 146.86, 127.13, 125.20, 118.88, 111.81, 111.23, 101.42, 55.79, 55.69. IR 3115 cm^−1^, 2943 cm^−1^, 2836 cm^−1^, 1615 cm^−1^, 1583 cm^−1^, 1546 cm^−1^, 1509 cm^−1^, 1267 cm^−1^, 1159 cm^−1^, 1140 cm^−1^, 967 cm^−1^, 821 cm^−1^, 604 cm^−1^. MS: *m*/*z* = 444 *m*/*z* calc = 444.23 for C_23_H_23_BF_2_O_6_.

Synthesis of ligand dimetoxycurcumin: the removal of the BF_2_ group was carried out using methanol and alumina under similar experimental conditions as previously reported [55].

DiMeOC, **3**: (1E,4Z,6E)-1,7-bis(3,4-dimethoxyphenyl)-5-hydroxyhepta-1,4,6-trien-3-one: yield 90%, orange powder, m.p. = 130 °C. ^1^H NMR (400 MHz, DMSO-*d*_6_) δ 7.59 (d, *J* = 15.9 Hz, 2H), 7.35 (d, *J* = 2.0 Hz, 2H), 7.26 (dd, *J* = 8.6, 2.0 Hz, 2H), 7.03–6.98 (m, 2H), 6.82 (d, *J* = 15.9, 2H), 6.10 (s, 1H), 3.83 (s, 6H), 3.80 (s, 6H). ^13^C NMR (100 MHz, DMSO-*d*_6_) δ 183.19, 150.96, 149.02, 140.40, 127.56, 122.90, 122.03, 111.67, 110.47, 100.99, 55.57. The IR spectrum and MS data are the same as previously reported and coincide with the molecular formula C_23_H_24_O_6_.

The synthesis of the magnesium metal complex was carried out as follows [22]: 1 mmol of DiMeOC was dissolved in 15 mL of EtAcO; later, 0.6 mmol of magnesium(II) acetate 4H_2_O dissolved in MeOH was added dropwise. The reaction mixture was stirred for 24 h. A yellow fine powder was filtered off in vacuo and washed with H_2_O.

DiMeOC-Mg, **4**: magnesium (1E,3Z,6E)-1,7-bis(3,4-dimethoxyphenyl)-5-oxohepta-1,3,6-trien-3-olate: yield 80%, yellow powder, m.p. = 230 °C. ^1^H NMR (400 MHz, DMSO-*d*_6_) δ 7.36 (d, *J* = 15.6 Hz, 2H), 7.25 (s, 2H), 7.13 (d, *J* = 8.3 Hz, 2H), 6.95 (d, *J* = 8.4 Hz, 2H), 6.71 (d, *J* = 15.6 Hz, 2H), 5.64 (s, 1H), 3.81 (s, 6H), 3.78 (s, 6H). ^13^C NMR (100 MHz, DMSO-*d*_6_) δ 181.24, 149.74, 148.99, 135.81, 128.77, 121.44, 111.72, 109.82, 103.06, 55.54, 55.51. IR 3349 cm^−1^, 2934 cm^−1^, 2833 cm^−1^, 1627 cm^−1^, 1598 cm^−1^, 1581 cm^−1^, 1544 cm^−1^, 1507 cm^−1^, 1439 cm^−1^, 1255 cm^−1^, 1024 cm^−1^, 809 cm^−1^, 470 cm^−1^. MS-MALDI-TOF: *m*/*z* = 817.118 *m*/*z* calc = 817.29 for C_46_H_46_MgO_12_.

The inclusion complex was prepared via the coprecipitation method [52,57] using the following procedure: in a 250 mL round flask 1 mmol of beta-cyclodextrin (BCD) was dissolved in 100 mL of distilled water; later, 1 mmol of magnesium metal complex (DiMeOC-Mg) dissolved in 100 mL of acetone was added dropwise, and the mixture was left in continuous stirring at room temperature for 12 h. A yellow precipitate formed was filtered off and washed with distilled water/acetone 1:1 to remove residual BCD or DiMeOC-Mg; the precipitate was dried in high vacuum.

DiMeOC-Mg-BCD, **5** (inclusion complex), yield 75%, pale yellow powder, m.p. = 320 °C. ^1^H NMR (400 MHz, DMSO-*d*_6_) δ 7.34 (d, *J* = 15.6 Hz, 4H), 7.25 (d, *J* = 2.0 Hz, 4H), 7.13 (dd, *J* = 8.4, 1.9 Hz, 4H), 6.95 (d, *J* = 8.4 Hz, 4H), 6.70 (d, *J* = 15.6 Hz, 4H), 5.74 (s, 7H), 5.69 (s, 7H), 5.63 (s, 2H), 4.83 (d, *J* = 3.6 Hz, 7H), 4.46 (t, *J* = 5.6 Hz, 7H), 3.81 (s, 12H), 3.78 (s, 12H), 3.64 (M, 14H), 3.57 (m, 7H), 3.34 (m, 21H). ^13^C NMR (100 MHz, DMSO-*d*_6_) δ 181.22, 149.73, 148.98, 135.79, 128.76, 121.45, 111.73, 109.81, 103.01, 101.95, 81.55, 73.06, 72.43, 72.04, 59.92, 55.54, 55.50. IR 3386 cm^−1^, 2930 cm^−1^, 2836 cm^−1^, 1628 cm^−1^, 1600 cm^−1^, 1582 cm^−1^, 1550 cm^−1^, 1510 cm^−1^, 1438 cm^−1^, 1420 cm^−1^, 1264 cm^−1^, 1157 cm^−1^, 1139 cm^−1^, 1026 cm^−1^. MS-ESI: *m*/*z* = 1949.8 *m*/*z* calc = 1950.15 for C_88_H_116_MgO_47_.

## 5. Conclusions

A rational design was undertaken based on the following three simple concepts: (a) synthesis of curcuminoids, (b) synthesis of the corresponding metal complexes, and (c) preparation of their inclusion complexes to render a more soluble, stable, and bioavailable formulation to optimize the cytotoxic and antitumoral effects. 

The DiMeOC-Mg-BCD formulation contributes a promising result that could enhance the effectiveness of conventional chemotherapy, including radiation therapy, and even reduce the associated side effects and increase the overall efficacy as an antitumoral agent.

DiMeOC-Mg and DiMeOC-Mg-BCD had antitumoral activity against the MDA-MB-231 cell line. DiMeOC-Mg at a dose of 1.5 mg/kg reduced tumor weight by 16.5%, while DiMeOC-Mg-BCD at 0.065 and 0.65 mg/kg reduced tumor weight by 29.45% and 33.76%, respectively, in chicken embryos. Furthermore, none of the compounds tested induced embryotoxicity (abnormalities or death in ovo). DiMeOC-Mg-BCD has excellent potential to be used as a therapeutic agent in the battle against breast cancer.

The present study strengthens the importance of formulating antitumor inclusion complexes from metal complexes of curcuminoids and may help expand the search for new formulations aimed at treating different types of malignant tumors and relieving human ailments.

## 6. Patents

An application for a patent is underway in the country of the authors.

## Data Availability

Data is contained within the article and Appendix A.

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
