# Peer review of "The Antitumoral Effect In Ovo of a New Inclusion Complex from Dimethoxycurcumin with Magnesium and Beta-Cyclodextrin"

_ijms, 2024, doi:10.3390/ijms25084380_

Round 1

Reviewer 1 Report

Comments and Suggestions for Authors

The research is original and important in the field of plant metabolites. The interpretation of the results by the statistical methods used are relevant. The methods is clear and well organized. The authors are to be commended for putting together an easily readable manuscript that is well put together and well written. This study has certain reference value. Although the results are interesting and their conclusions are valid.

Just some recommendations regarding:

the clarity/resolution of Figures 6, S18 and S19.

elaborate the solubility study and results

Minor editing errors, for instance at:

line 255: ( see Figure 3)

line 309: (approx. 160 degrees, Figure 5 c))

Author Response

Dear Referee 1

Please find our response to your valuable suggestions.

Thank you for your valuable time.

Reviewer 2 Report

Comments and Suggestions for Authors

The article written by Marco A. Obregón-Mendoza is an interesting research that attempts to provide a new therapeutic solution for the treatment of breast cancer. I believe the research was well conducted but requires some improvements:

The Introduction should be complemented by a figure illustrating the chemical structures of curcumin and its analogs, outlining their mechanism of antitumor action.

Authors would benefit from elaborating on the aspects presented in lines 68-73, providing some detail on the mechanisms of antitumor action without relying heavily on accumulated references. Optionally, the data could be presented in a table format.

I observed that an important work in the studied research area ("Sohail M - A Promising Anticancer Agent Dimethoxycurcumin: Aspects of Pharmacokinetics, Efficacy, Mechanism, and Nanoformulation for Drug Delivery. Front Pharmacol. 2021, doi: 10.3389/fphar.2021.665387") was not considered by the authors in supporting their argumentation, instead referring to numerous personal works. I believe a research is all the more significant when it acknowledges prominent works in the field, reflecting the scientific community's interest in the topic.

In the Materials and Methods section, the authors should provide the manufacturer's name and country of origin for all substances and equipment used. It would have been beneficial if the authors had maintained the presentation in this section according to the subsections from the Discussion section.

Minor observation:

Use a space before the sqare bracket with reference number.

Author Response

Dear Reviewer 2

We appreciate your valuable suggestions and time.

Truly yours

Raul G Enriquez

Reviewer 3 Report

Comments and Suggestions for Authors

The article is devoted to synthesis of new inclusion complex containing dimethoxycurcumin with magnesium and beta-cyclodextrin. The article is written logically, the direction of the search for new biologically active agents is relevant and modern. The conclusions made by authors are consistent with the study. The paper can be accepted for publication after minor revisions.

Introduction

Would be better if authors provide the figure with mentioned curcuminoid (lines 64-65)

Results

• lines 105-112: The synthetic route is presented unclear. The 1st step is not provided in the scheme 1, so it would be better if authors will put additional step or will rewrite the sentence, and mention for what purpose used 2,4-pentanedione and BF3·THF synthon

• "Condensation of 2 moles of 3,4-dimethoxybenzaldehyde" should be corrected. It seems that 2 moles of 3,4-dimethoxybenzaldehyde reacted between each other. Change “of” to “with” and add the compounds which it was reacted with.

• Add more detailed information in the synthesis

- What is the role of tributyl borate and butylamine in 1st step, which solvent was used, time conditions etc.

Add information in the results part which exactly type of inclusion complex was obtained, if any.

Please check all text for italics (for example line 103), bold numbering (for example line 109), superscripts (for example line 253) subscripts, spaces (for example line 124, 256 and before ref brackets)

Comments on the Quality of English Language

Minor editing of English language required

Author Response

Dear Reviewer 3:

On behalf of my colleagues, we express our gratitude for your valuable time and suggestions

Truly yours

Raul G Enriquez

Reviewer 4 Report

Comments and Suggestions for Authors

The manuscript "The antitumoral effect in ovo of a new inclusion complex from dimethoxycurcumin with magnesium and beta-cyclodextrin" is devoted to the synthesis and study of the biological activity of an anticancer agent based on curcumin and magnesium ion. This is a very relevant research topic. The fact is that breast cancer, especially triple negative according to morphological classification, is extremely difficult to treat. Moreover, today there are no targeted drugs (I mean monoclonal antibodies) to treat this form of the disease. Therefore, the creation of drugs that counteract this form of the disease is a very urgent and necessary task. Indeed, the synthesized compound is very effective against cancer cells and the biological part of the study is also very impressive. The only drawback in this article is the conclusions about the activation of certain signaling pathways based on the analysis of RNA expression. The fact is that sometimes the presence of high levels of one or another m-RNA does not at all indicate an increase in one or another protein. That is, the synthesized substance is indeed an inducer of apoptosis in cancer cells, but it is not at all necessary that this compound activates the signaling pathways described in this article. At a minimum, a Western blot of the described proteins is required, and only then can we judge the activation or induction of intracellular signaling. We cannot draw such far-reaching conclusions solely based on the analysis of m-RNA with gene-specific primers.

I advise the authors to think over and slightly modify the description of the results.

The article deserves to be published in this scientific publication.

Author Response

Dear Reviewer 4:

On behalf of our research group, I express my gratitute for your valuable suggestions and time.

Truly yours

Raul G. Enriquez

Round 2

Reviewer 1 Report

Comments and Suggestions for Authors

Agree with the revised manuscript